# Tuberculous Meningitis in Children: Reducing the Burden of Death and Disability

**DOI:** 10.3390/pathogens11010038

**Published:** 2021-12-30

**Authors:** Julie Huynh, Yara-Natalie Abo, Karen du Preez, Regan Solomons, Kelly E Dooley, James A Seddon

**Affiliations:** 1Oxford University Clinical Research Unit, Centre for Tropical Medicine, Hospital for Tropical Diseases, Ho Chi Minh City 700000, Vietnam; 2Centre for Tropical Medicine and Global Health, Nuffield Department of Medicine, Oxford University, Oxford OX1 2JD, UK; 3Infectious Diseases Unit, The Royal Children’s Hospital Melbourne, Parkville, VIC 3052, Australia; Yara-Natalie.Abo@rch.org.au; 4Tropical Diseases Research Group, Murdoch Children’s Research Institute, Melbourne, VIC 3052, Australia; 5Desmond Tutu TB Centre, Department of Paediatrics and Child Health, Stellenbosch University, Cape Town 7600, South Africa; karen_dupreez@sun.ac.za (K.d.P.); james.seddon@imperial.ac.uk (J.A.S.); 6Department of Paediatrics and Child Health, Faculty of Medicine and Health Sciences, Stellenbosch University, Cape Town 7600, South Africa; regan@sun.ac.za; 7Department of Medicine–Infectious Diseases, Johns Hopkins University School of Medicine, Baltimore, MD 21218, USA; kdooley1@jhmi.edu; 8Department of Infectious Diseases, Imperial College London, London SW7 2BX, UK

**Keywords:** tuberculous meningitis, TBM, disseminated, central nervous system

## Abstract

Tuberculous meningitis disproportionately affects young children. As the most devastating form of tuberculosis, it is associated with unacceptably high rates of mortality and morbidity even if treated. Challenging to diagnose and treat, tuberculous meningitis commonly causes long-term neurodisability in those who do survive. There remains an urgent need for strengthened surveillance, improved rapid diagnostics technology, optimised anti-tuberculosis drug therapy, investigation of new host-directed therapy, and further research on long-term functional and neurodevelopmental outcomes to allow targeted intervention. This review focuses on the neglected field of paediatric tuberculous meningitis and bridges current clinical gaps with research questions to improve outcomes from this crippling disease.

## 1. Introduction

Young children and individuals living with HIV are at high risk of progressing to tuberculosis (TB) disease following TB infection and are at elevated risk of progressing to severe forms of disease such as disseminated TB and tuberculous meningitis (TBM) [1]. TBM is the most devastating form of TB and is associated with high mortality and morbidity. Untreated, all children will die [2]. Even if diagnosed and treated, 20% of children die and of those surviving over half have neurological disability [3]. TBM in children, therefore, merits special consideration. In this article we review the natural history and pathogenesis of TBM in children, the epidemiology of the disease, approaches to diagnosis, developments in treatment and considerations for long-term prognosis. We present recent research findings and areas that require prioritised future investigation (Table 1).

## 2. Natural History and Pathogenesis

After inhalation of *M. tuberculosis*-containing droplets, bacilli may deposit into the terminal alveoli. If they overcome the structural defences and innate immune response, an inflammatory process involving cytokine release, granuloma formation, and primary infection will ensue in the lungs. During this process a bacteraemia can occur where bacilli are filtered into draining lymph nodes and then onto the systemic circulation and distant sites, including the central nervous system (CNS) [4]. *M. tuberculosis* may then invade the blood brain barrier via (1) rearrangement of actin; (2) *M. tuberculosis* virulence factor(s) interacting with extracellular brain endothelium factors to facilitate bacillary endothelial adhesion; or (3) the ‘Trojan horse’ mechanism via infected macrophages and neutrophils [5,6,7,8]. Once bacilli have gained access to the brain, a subcortical or meningeal ‘Rich focus’ is formed via activation of microglial cells and astrocytes [9,10]. When this Rich focus is activated (rapidly in the context of miliary TB, as demonstrated in young children [10], or months to years later), *M. tuberculosis* is released into the subarachnoid space triggering a T-cell mediated inflammatory cascade including induction of pro- and anti-inflammatory cytokines such as tumour necrosis factor-alpha, interferon-gamma, interleukin (IL) 1b, IL-6, IL-8, and IL-10 [4]. The consequent formation of exudate envelopes arteries and nerves, disrupting cerebrospinal fluid (CSF) flow and contributing to development of vasculitis in the vessels of the Circle of Willis, the vertebrobasilar system, and the perforating branches of the middle cerebral artery. Resultant hydrocephalus and infarct contribute to the clinical presentation in TBM [11].

Whether the mycobacteria are contained or cause clinical disease, and the extent of clinical disease, is determined by an interplay of host immune response and *M. tuberculosis* virulence factors, however our understanding of these processes remains incomplete. Studies in paediatric [12,13,14] and adult TBM [13] demonstrate associations between immune mediators and clinical outcome, and suggest that a disequilibrium of pro- and anti-inflammatory cytokines underlies the severity and course of TBM. This balance can be regulated by Leukotriene A4 Hydroxylase (LTA4H); a gene that encodes an enzyme which influences the balance of pro- and anti-inflammatory eicosanoids seen in intracerebral inflammation. Variations of the LTA4H genotype may contribute to heterogeneity of the inflammatory response and outcomes in TBM [15] Studies are ongoing to further examine the role of the LTA4H genotype on the immunoinflammatory response and the possibility of personalising adjunctive anti-inflammatory therapy based on host genotype [13] Clues to further understanding the biology of cerebral injury in TBM have come from biomarker signatures in TBM-infected children presenting with stroke [1,14] and transcriptional profiles demonstrating compartmentalisation of the immune response within the CNS (ventricular vs. lumbar CSF) [16].

A review of the natural history of childhood intrathoracic tuberculosis in the pre-chemotherapy era found that young age (<2 years of age) was the major determinant of progression from TB infection to disease; pulmonary disease developed in 30–40% and TBM or miliary disease in 10–20% of infants, and the highest risk was within 4 months of infection [1]. In a world-wide meta-analysis of case-control studies, neonatal Bacille Calmette-Guérin (BCG) vaccination was shown to protect against TBM in children up to 5 years of age with a pooled efficacy of 73% [17,18]. Non-specific symptoms and the accompanying difficulty with early diagnosis highlight the importance of TBM prevention with BCG vaccination in young children. TB preventative therapy following TB exposure is another key preventive strategy that substantially lowers the risk of developing TB disease in young children [19] but implementation is poor in high TB burden settings [20].

## 3. Epidemiology

The World Health Organization (WHO) estimates that 1.2 million children (<15 years of age) developed TB in 2019 [21]. Yet, only 523,000 of these were notified by TB programs globally that year, leaving more than half of all children with TB undiagnosed or diagnosed but not reported [21]. Despite this large reporting gap, substantial progress has been made to strengthen paediatric TB surveillance since 2011, including reporting of age-disaggregated data on case notifications in 5-year age bands and also on outcomes for both drug-susceptible and drug-resistant paediatric TB since 2020 [21]. Despite current standard treatment and prevention strategies, the large number of children with undiagnosed TB (case detection gap) results in TB remaining one of the top 10 causes of childhood mortality [22].

Childhood TB accounts for 5–20% of the total TB caseload in a population, depending on the population age structure, TB and HIV prevalence, and availability of preventative measures [23]. Most low- and middle-income countries, typically those also affected most by TB, have a relatively young population (broad-based population pyramid) [24]. In these countries, the annual risk of TB infection is high, with younger children having an increased risk of TB exposure and infection. As the risk for progression from TB infection to TB disease, and even more so to TBM, is age-related and disproportionately high in children younger than 2 years, more TB infections in this age group not only leads to more children who develop TB, but also to more children with TBM [1].

At present, there are no estimates for the number of children affected by TBM. Current TB surveillance data does not require reporting of severe forms of TB, such as TBM, and even the number of children diagnosed, treated and reported are not known. This limits our ability to raise an adequate and effective response to TBM in children [25]. A study from Germany demonstrated that although the overall proportion of TB cases that were TBM was ~1%, this figure was 3.9% in children <5 years, 2.2% in children aged 5–9 years and 1.3% in children aged 10–14 years [26]. If 2% of childhood TB is TBM, then 20,000 childhood TBM cases are to be estimated each year globally. Including surveillance of TBM as part of monitoring and evaluation of paediatric TB care can help us measure and report both on the burden and outcomes of TBM in children and identify health system challenges. Given the substantial morbidity and often life-long disability suffered by children who survive, data on the burden of post-TBM health in children is critical to ensure adequate healthcare support is available to these children and their families. One way of better understanding the relationship between TBM in children and the health system response, is through cascade analysis. By evaluating drop offs at each stage in the care cascade, programmatic challenges and research priorities can be identified (Figure 1). There is a sequential drop-off in numbers of TBM-infected children at each stage of care from presentation to health services to long-term outpatient monitoring. Combined with the real-world challenges at each of these stages, the true number of children with TBM who die or survive with sequelae is far greater than currently appreciated.

## 4. Diagnosis 

TBM in paediatric practice is diagnosed based on a combination of clinical, laboratory and neuroimaging findings (Figure 2) [27]. Clinical prediction rules to distinguish TBM from other forms of meningitis have been proposed [28,29], whilst diagnostic algorithms such as the uniform research case definition [30] are designed for research purposes only. Clinical diagnostic rules are hampered by variable performance in different settings and lack of external validation [31]. To improve outcomes clinicians should maintain clinical suspicion and empirically treat suspected TBM without waiting for confirmatory results.

TBM can affect patients of all ages, however the brunt of disease is felt in early childhood with peak incidence commonly between 2 to 4 years [32] when the brain is still developing. It is difficult to make an early clinical diagnosis of TBM in childhood, subsequently resulting in delayed diagnosis and treatment, with an often inevitable poor outcome [3,32,33]. The classical presentation of TBM is as a subacute meningitic illness, however neck stiffness is often absent early in the course of the disease [34]. In order to recognize early stage TBM in children, clinicians in TB-endemic settings must be aware that TBM most commonly presents with non-specific symptoms of general ill health. However, the persistence of symptoms allows differentiation from other common illnesses with similar presentation (e.g., influenza) [35]. Household exposure to an adult source case with pulmonary TB within the previous year should heighten suspicion of TBM. 

Both neuroimaging and CSF analysis are essential in the diagnostic assessment of paediatric TBM (Figure 2). Computed tomography (CT) is more readily available than magnetic resonance imaging (MRI) in resource-constrained settings [36]. Classic CT findings include pre- and post-contrast basal meningeal exudates, hydrocephalus, infarcts and tuberculomas [37]. MRI is superior in the early identification of TBM as gadolinium enhancement can detect small leptomeningeal tuberculomas, and diffusion-weighting can detect early infarction, both not reflected on CT [38,39]. Leucocytosis with lymphocyte predominance, elevated protein and abnormally decreased CSF glucose are highly suggestive of TBM [32,40,41]. Both an absolute CSF glucose value of <2.2 mmol/L and CSF protein >1 g/L differentiate TBM from viral or no meningitis in children with good specificity, albeit with poor sensitivity. CSF to serum glucose ratio, infrequently performed, is essential to provide additional information value [42].

Mycobacterial confirmation in children with presumed TBM is difficult due to low CSF volumes obtained and the paucibacillary nature of TBM. CSF microscopy is hampered by low sensitivity [43] however yields may be improved by centrifugation and longer examination time. Even though sensitivity of CSF culture is higher than microscopy, it remains sub-optimal and the result rarely influences clinical management due to delays of up to 8 weeks [31]. Nucleic-acid amplification testing offers the prospect of a rapid and specific result, however few tests have undergone validation. Xpert MTB/ RIF (Cepheid, Sunnyvale, CA, USA) is a commercial, real-time PCR-based assay for the detection of *M. tuberculosis* in clinical specimens. In 2015 Xpert MTB/RIF has been recommended by the WHO as an essential diagnostic test if TBM is suspected, however caution is advised for its use as a ‘rule out’ test [44,45]. The second generation Xpert MTB/RIF, XpertUltra, detects TBM with marginally higher sensitivity than Xpert and poor negative predictive value in adults, meaning it cannot be used to rule out TBM [46,47,48]. In children, poor positive predictive value may be more of an issue. The meaning of trace positive results on XpertUltra also is not understood. Inspite of these limitations, Xpert MTB/Rif or Xpert Ultra is able to provide a result in under 1.5 h; a crucial advantage when initiating early anti-tuberculosis therapy if positive (Figure 2). We propose a diagnostic algorithm for presumptive TBM in children which incorporates clinical, CSF, neuroimaging features and rapid diagnostic results and provide guidance for possible clinical scenarios, including when diagnostic tools are negative or inconclusive for TBM and the clinician left to make a clinical judgement (Figure 2).

Diagnostic tests for TBM are relatively expensive and inaccessible in resource-constrained areas, invasive and perform poorly in isolation [30,49]. Recent technological advances have made it possible to screen for many biomarkers in a minute volume of CSF. A three-marker CSF biosignature comprising IL-13, VEGF and cathelicidin LL-37, diagnosed childhood TBM with a sensitivity of 52%, specificity of 95%, with positive and negative predictive values of 91% and 66% respectively [50]. Validation of this three-marker CSF biosignature in a different cohort revealed positive and negative predictive values of 90% and 59.5% respectively [49]. In a study investigating potentially useful host CSF biomarkers in childhood TBM, a combination of IFN-γ, MPO and VEGF showed good accuracy (AUC = 0.97, up to 91.3% sensitivity and up to 100% specificity) [49]. Low CSF tryptophan concentration is associated with survival in TBM patients [51]. The CSF metabolome in TBM is also characterized by amino acids (besides for tryptophan), organic acids, nucleotides and carbohydrates, all linked to altered neuro-energetics [52,53,54]. CSF metabolomics studies of paediatric TBM [55,56,57] is advancing from a proof-of-concept, exploratory phase towards validation and standardization as biomarkers [54]. Other host-immune response testing (e.g., CSF IFN-γ release assays) have only moderate diagnostic accuracy [58] while systematic reviews on CSF adenosine deaminase levels concluded that heterogeneity in methods and data limit applicability in clinical practice [31,59,60]. It remains to be seen whether biomarker-based approaches can be transformed into easy-to-use point-of-care diagnostic tests [61], especially, in resource-limited settings, but it is likely that the future of TBM diagnosis will require a combination of pathogen testing and host-immune biomarkers.

## 5. Treatment

Currently, the WHO recommends treatment for paediatric TBM similar to that used for pulmonary TB—isoniazid, rifampicin, and pyrazinamide plus ethambutol—given at standard doses, albeit with treatment extension from 6 to 12 months. This recommendation was provided with low quality of evidence prior to the implementation of GRADE procedures in WHO guidelines processes [62]. However, to be effective, drugs must be present at therapeutic concentrations at the site of disease. This is critically important early in treatment, to prevent mortality and neurocognitive disabilities. The latter is particularly important for children whose developing brains render them susceptible to brain injury from TBM or its treatment. Several first-line anti-tuberculosis drugs (e.g., rifampicin and ethambutol) have poor penetration across the blood-brain and the blood-CSF barriers [63]. Rifampicin remains an essential medication for TBM [64] but standard dosing yields CSF concentrations that are below the minimal inhibitory concentration (MIC) for *M. tuberculosis* in a large proportion of patients, including children [65,66,67]. A recent model-based meta-analysis using emerging data from adult trials showed a strong positive correlation between rifampicin concentrations and survival, with doses of at least 30 mg/kg predicted to improve survival substantially compared to 10 mg/kg [68]. A dose of 35 mg/kg daily is being tested in a definitive Phase 3 trial [69] but higher mg/kg doses than that will be needed to achieve adult-equivalent exposures in children [70]. In children, a small trial suggested that a dose increase to 30 mg/kg improved neurocognitive function compared to standard dosing [71].

While higher rifampicin exposures were associated with improved outcomes in Indonesia for adults, in a large adult study in Vietnam, isoniazid exposures correlated more strongly with survival [72]. While isoniazid penetrates freely into CSF, patients with fast n-acetyltransferase 2 (NAT2) genotype may experience sub-therapeutic concentrations. Pyrazinamide passes easily into CSF, but its contribution to TBM treatment remains poorly-defined [73]. Recently approved as an alternative to standard of care, the ‘Cape Town regimen’ takes into account the pharmacokinetics of anti-TB drugs, including information about their CNS distribution. This regimen, used for paediatric TBM in South Africa for over 20 years, includes high-dose rifampicin, high-dose isoniazid, ethionamide, and pyrazinamide for 6 months [74]. Following a recent systematic review and meta-analysis (3 studies, 1006 participants) showing improved mortality in children who received a 6-month intensified regimen for TBM compared to the standard 12 months, WHO issued a rapid communication on TB management suggesting that the shortened regimen could be used as an alternative in children and adolescents [75]. Two paediatric clinical trials SURE (ISRCTN40829906) and TBM-KIDS (NCT02958709) are currently underway and will measure mortality, functional status and neurocognitive outcomes in children receiving shortened intensified drug regimen (optimised dosing of rifampicin, with isoniazid, pyrazinamide, and levofloxacin) versus the standard WHO regimen.

Second-line drugs may be needed either for drug-resistant TBM, or to replace drugs that display poor CNS penetration or cause toxicity. Fluoroquinolones achieve high CSF concentrations and are generally safe in children, though there have been some reports of intracranial hypertension and seizures associated with these agents [76,77]. Linezolid is used for CNS infections involving gram positive bacteria, and early data from cohort studies suggest this drug is also good for TBM [78,79,80] though its toxicities make use beyond 8 weeks challenging [81]. Delamanid or pretomanid may be useful in TBM [82] and bedaquiline appears to have similar free-drug concentrations in CSF as in plasma [83], but the place of these newer anti-TB drugs in TBM remains to be established. Another clinically challenging situation is HIV-associated TBM. While in adults, antiretroviral therapy (ART) is typically delayed until the intensive phase of TBM treatment is complete [84] the right timing for introduction of ART in children, who tend to have more rapid progression to severe HIV, is unknown. 

To address the pathological impact of the host response, adjunctive steroids are given as standard of care in TBM. Additional host-directed therapies may be useful for select patients, including aspirin or low-dose thalidomide [85,86,87]. Whilst RCTs evaluating aspirin are underway, the role of non-corticosteroids adjuvant agents is not yet established. Management of CNS complications of TBM - infarcts, seizures, hydrocephalus - remains central to the care of individuals with this disease, and some published guidance documents have been produced to aid clinicians [88,89].

## 6. Outcomes

Despite the advent of anti-TB chemotherapy and corticosteroids most deaths (95%) from TBM will still occur by 6 months [69]. An observational report in 1961 on long-term outcomes in children with TBM, was the first to highlight persistent neurodevelopmental sequelae years following completion of TB treatment [90]. Younger age (<2 years of age) and more severe disease at diagnosis was associated with worse neurological sequelae [90]. Whilst there has been recent progress in management of TBM, it has not yet translated into outcome benefits in clinical practice. Adjuvant corticosteroids reduce mortality in TBM, but they do not reduce neurodisability in survivors [91].

A concerning 65% of children who survive TBM do so with some form of disability ranging from motor, sensory, cognitive to developmental deficits [3,92]. This is likely to be under-identified due to lack of surveillance and a standardised approach to follow-up. Reported outcomes vary depending on prevalence of risk factors including HIV co-infection, drug resistance [93] severe hydrocephalus, cerebral infarct, brainstem dysfunction, raised intracranial pressure and malnutrition [32,33,94,95,96]. Lack of access to supportive care and neurorehabilitation in low-resourced, TB-endemic settings further exacerbate poor outcomes [97]. Whilst reasons for poor outcomes are likely to be multifactorial, diagnosis before the onset of coma remains the most crucial factor predicting survival and favourable outcomes [98].

Predicting prognosis in children with TBM is difficult owing to its insidious onset, diversity of immunopathology under various genetic influences and limited knowledge about the durability of brain injury in children. Emerging evidence, using biomarkers of cerebral injury usually seen in neurodegenerative disease, suggest that brain injury in TBM increases over time and lasts long after completion of TB treatment and corticosteroids [12]. New molecular techniques (i.e., transcriptomics, proteomic and metabolomics) characterising signatures linking clinical phenotypes with TBM outcomes will likely advance knowledge in markers of prognostication [16,51]. Currently, the best association with 6-month neurological outcomes is clinical staging of severity using refined MRC grading system one week after diagnosis [99].

Long-term outcomes in children with TBM include cerebral palsy (e.g., hemiplegia) vision impairment (e.g., blindness), hearing loss, cognitive impairment (e.g., learning capacity), chronic seizure disorder, behavioural disturbance (e.g., Attention Deficit Hyperreactivity Disorder) and developmental disability [32,100]. Notably, neurocognitive deficits can occur without accompanying physical disability [33]. Neurocognitive and functional impairment is difficult to fully characterise due to diversity of phenotypes and recovery across age groups, loss to follow-up, and a lack of testing specifically developed or validated for evaluation in TBM [101].

## 7. Conclusions

TBM in children remains a devastating disease, associated with substantial morbidity and mortality. It is challenging to diagnose and treat early; much of the damage has already occurred before the child is started on appropriate therapy. Our understanding of the pathophysiology and epidemiology of the condition is improving, and novel diagnostic approaches are being developed. New anti-TB drug regimens and dosing strategies are under evaluation, emerging host-directed therapies are being explored and supportive care is improving. However, substantial gains could be made by strengthening existing health systems, to allow earlier diagnosis and appropriate treatment and optimised surveillance

## Figures and Tables

**Figure 1 pathogens-11-00038-f001:**
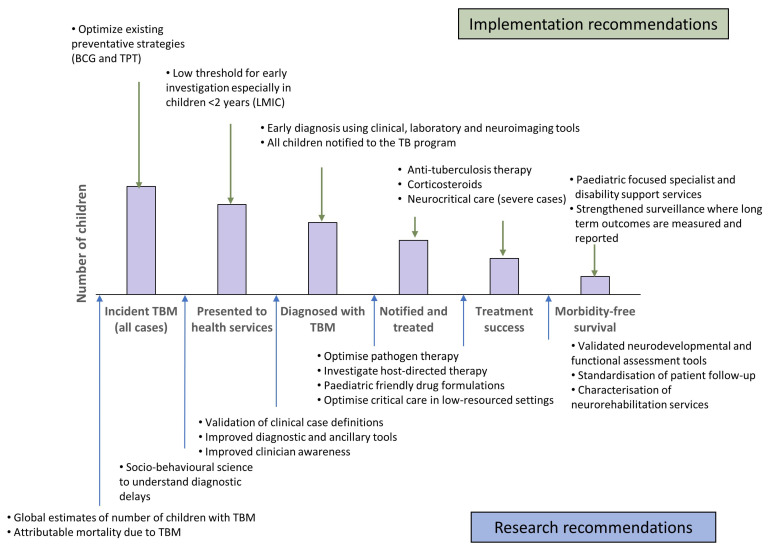
The cascade of care for children with tuberculous meningitis, illustrating how new research developments and optimised programmatic care could reduce drop off at each stage. BCG: bacille Calmette–Guérin; TPT: tuberculosis preventive therapy; LMIC: low- and middle-income country; TBM: tuberculous meningitis; TB: tuberculosis.

**Figure 2 pathogens-11-00038-f002:**
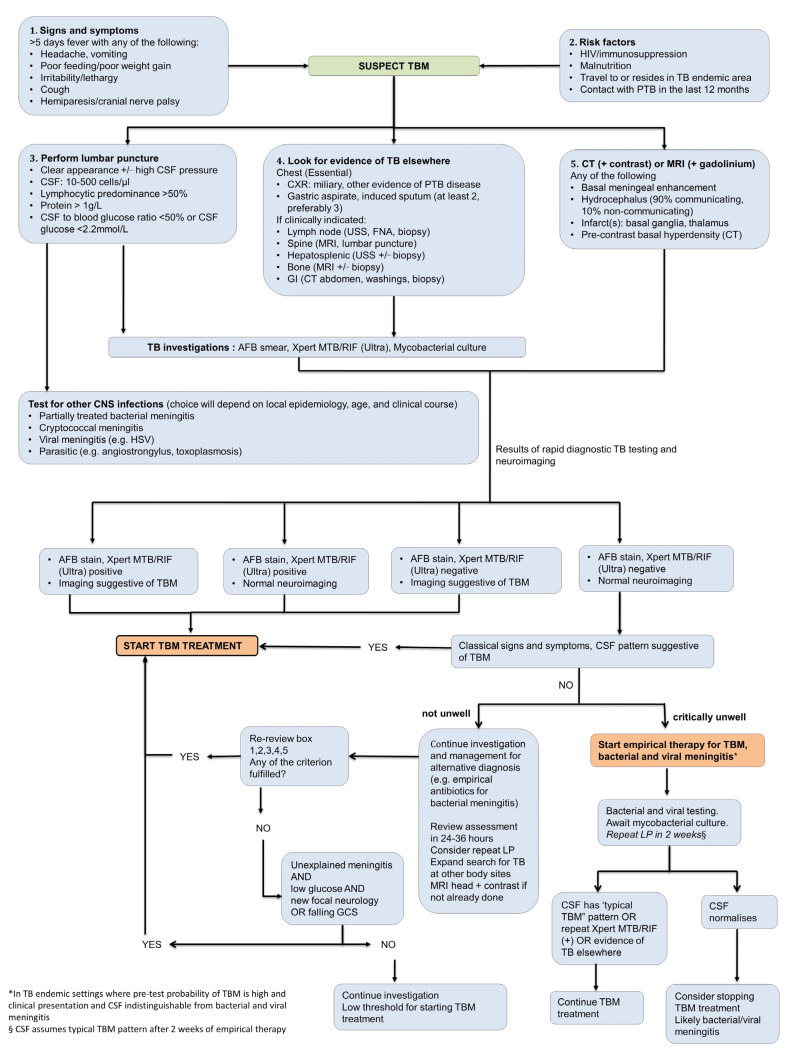
A proposed diagnostic algorithm for treatment decision making in children with presumptive tuberculous meningitis. Xpert MTB/RIF or Xpert Ultra is recommended by WHO as the initial diagnostic test in CSF for TBM rather than smear microscopy/culture although evidence for test accuracy of Xpert Ultra for TBM in children is limited. TBM: tuberculous meningitis; TB: tuberculosis; PTB: pulmonary TB; CSF: cerebrospinal fluid; CXR: chest X-ray; USS: ultrasound scan; FNA: fine needle aspiration; MRI: magnetic resonance imaging; GI: gastrointestinal; CT: computerised tomography; AFB: acid fast bacilli; CNS: central nervous system; HSV: herpes simplex virus; LP: lumbar puncture; GCS: Glasgow Coma Scale score.

**Table 1 pathogens-11-00038-t001:** Current knowledge and research gaps for childhood tuberculous meningitis.

Research Area	Current Knowledge	Research Gaps
Pathogenesis	Novel biomarker and host genotype studies offer new insight into TBM pathophysiology Young age is a major determinant of progression from TB infection to TBM BCG vaccination and TB preventive therapy for children exposed to TB cases are important preventative strategies	Further understanding of the mechanism of TB dissemination from lungs to CSF and CSF invasion Further understanding of host and pathogen factors that determine why some children develop TBM
Epidemiology	More than half of all children with TB globally are undiagnosed or unreported Infants have an up to 20% risk of developing TBM following TB infection Advanced disease stage at diagnosis is associated with high mortality and morbidity	The number of children with TBM are currently unknown, and modelling studies are needed to provide estimates of the burden, morbidity, and mortality in children globally Operational research can help to identify and mitigate the impact of drivers behind diagnostic delays and missed opportunities for prevention Improve mechanisms for reporting of TBM in all age groups to national TB surveillance programs
Diagnostics	TBM can be diagnosed with reasonable confidence with clinical, laboratory, and neuroimaging findings MRI is superior to CT imaging for children being evaluated for TBM, both from a diagnostic perspective but also to delineate pathological and prognostic features *M. tuberculosis* detection remains the ‘gold standard’ diagnostic test but is limited by poor sensitivity Biomarkers have potential to improve our ability to discriminate children with TBM from children with other causes for their symptoms and signs	Establish validated clinical case definitions in adults and children taking into account different settings stages of TBM and HIV-status Investigate the ability of MRI CSF flow imaging and thin slice CT to differentiate communicating and non-communicating hydrocephalus Investigate utility of other modalities (e.g., 18F-PET/CT) to identify early small infarctions missed with conventional imaging Develop new adequately sensitive, accessible, and rapid diagnostic tests, especially at point of care to allow prompt diagnosis Further investigate the role of non- or less invasive TB testing (e.g., serum, urine, or saliva) in diagnosing TBM in patients where CSF is difficult or cannot be obtained Further investigate the utility of new omic technology, transcriptional and metabolomic biomarkers in diagnosing TBM from other non-TBM CNS infections, including in various populations, stages of TBM, and HIV-status
Treatment	Current recommended doses of TB drugs to treat TBM do not reach optimal CSF levels Higher doses of rifampicin are required to penetrate the blood CSF barrier High CSF concentrations of isoniazid are associated with improved survival To date, the only adjuvant therapy proven to reduce mortality in TBM is corticosteroids for up to 8 weeks Neurocritical care in severe or complicated disease should be part of routine management of TBM	What is the optimal dose, drug regimen, and duration of TB drugs to adequately treat TBM? Is shortened therapy non-inferior to the standard 12 months? Should patients who are fast acetylators receive different doses of isoniazid than slow acetylators? Besides rifampicin, isoniazid, and pyrazinamide, which first or second-line TB drug is the most optimal 4th drug in a drug-susceptible TBM regimen? When is the optimal time to start TB treatment in HIV-positive children? Do the new second line TB drugs (bedaquiline, delamanid, and pretomanid) have a role in treating drug-resistant TBM? Can new host-direct therapy (e.g., high-dose aspirin, thalidomide, and monoclonal-antibodies such as TNF-alpha inhibitors) reduce mortality or neurodisability in children with TBM? What is the optimal management for CNS complications in TBM? (e.g., hydrocephalus, tuberculomas, and paradoxical reactions) What is the optimal supportive and critical care in low-resourced settings to improve mortality and morbidity?
Long term morbidity	Morbidity and mortality from TBM remain unacceptably high, even if treated Reasons for poor outcome are multifactorial; however, the most important predictor of poor outcome is diagnostic and treatment delay Although the array of long-term sequelae is broad, the most common long-term outcomes are physical and developmental disability Neurocognitive deficits can occur without physical disability and have important psychosocial and educational consequences for children, especially those with immature brains	Establish validated and culturally appropriate tools to assess neurodevelopment and function in children with TBM Standardisation of patient follow up and strengthened surveillance to include physical, neurocognitive, and neurodevelopmental assessments Evaluation of whether optimised anti-TB therapy and host-directed therapy could improve long-term neurodevelopmental and neurocognitive outcomes across various paediatric age ranges Evaluation and characterisation of early intervention and targeted neurorehabilitation services to improve long-term outcomes

TBM: tuberculous meningitis; TB: tuberculosis; BCG: bacille Calmette–Guérin; CSF: cerebrospinal fluid; MRI: magnetic resonance imaging; CT: computerised tomography; PET: positron emissions tomography; CNS: central nervous system; TNF: tumour necrosis factor.

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
