# Peer review of "Tuberculous Meningitis in Children: Reducing the Burden of Death and Disability"

_pathogens, 2021, doi:10.3390/pathogens11010038_

Round 1

Reviewer 1 Report

Dear Authors,

this was a comprehensive and easily reading review on an important public health issue with great socioeconomic implications. Thank you for this submission.

this is a comprehensive review on an important public health issue with great socioeconomic implications. The thematic sections, in which your manuscript has been divided (pathogenesis, epidemiology, diagnosis, treatment and outcome), promote the pleasant reading and understanding of the content even from non-experts in the field. The synopsis of current knowledge and research gaps is of high importance, and so are your research-implementation recommendations, suggesting several paths for future research on the subject. The proposed diagnostic algorithm for decision-making and treatment in children with presumptive tuberculous meningitis may become enormously useful for clinicians active in the field of infectious diseases in pediatric patients. For all the above mentioned reasons thank you for this submission and keep on providing qualitative scientific work.

Best Regards

Author Response

Thank you for your comments.

Reviewer 2 Report

The paper offers a review of the very important topic of meningeal tuberculosis in children. This presentation of tuberculosis is a particularly complicated clinical diagnosis, treatment that has high incidence of mortality. Children are at particular risk for this manifestation of the disease and are at higher risk for life long neurological disabilities specially in low and middle income countries. 

The authors aim to identify clinical gaps and propose research questions to improve the outcome of the disease, however fail to include some recent publications.

The article is well written and easy to follow. 

General Concept comments -

1 - The document is a literature review, however some recent reviews on the topic have been published that should be included in the discussion and referenced.

  J Pediatr Neurosci. 2018 Oct-Dec; 13(4): 373–382.

doi: 10.4103/JPN.JPN_78_18

   Indian J Med Res. 2019 Aug; 150(2): 117–130.

doi: 10.4103/ijmr.IJMR_786_17

Particularly,

  J Clin Microbiol. 2021 Mar; 59(3): e01771-20.

Tuberculous Meningitis: Pathogenesis, Immune Responses, Diagnostic Challenges, and the Potential of Biomarker-Based Approaches

Charles M. Manyelo,a Regan S. Solomons,b Gerhard Walzl,a and Novel N. Chegou a (Published online 2021 Feb 18. Prepublished online 2020 Oct 21. doi: 10.1128/JCM.01771-20)

This review focuses on many similar aspects of the authors current document and would enrich the discussion in the very least. Repeating same arguments should be avoided.

2 - Other algorithms for diagnosis have been reported, and are not referenced.

3 - The section of ‘Natural History and Pathogenesis’ lacks a more thorough explanation of the immune response during the infection.

4- Both figure 1 and 2 are very helpful for the discussion and interpretation of the authors key points. However, a more detailed examination in the text should be included. This is an area that distinguishes the paper and should be exploited and expanded

Minor issues -

1 - The authors correctly point out that prompt diagnosis and reporting are important factors in successful treatment of TBM, however no specific recommendation or research is proposed.

2 - In the text the authors correctly state that Xpert is not sensitive enough for TBM, but it is added in the Figure 2. Please explain.

If Xpert is not recommended, what would the authors suggest.

3- A key point would be to identify and validate biomarkers, as the authors state in the text.

Are there any genetic predispositions for TBM?

How does malnutrition factor into the prognosis?

Is there an estimate on how long between pulmonary TB infection and TBM development?

Author Response

General Concept comments -

1 - The document is a literature review, however some recent reviews on the topic have been published that should be included in the discussion and referenced.

  J Pediatr Neurosci. 2018 Oct-Dec; 13(4): 373–382. doi: 10.4103/JPN.JPN_78_18

   Indian J Med Res. 2019 Aug; 150(2): 117–130. doi: 10.4103/ijmr.IJMR_786_17

Particularly,

  J Clin Microbiol. 2021 Mar; 59(3): e01771-20. Tuberculous Meningitis: Pathogenesis, Immune Responses, Diagnostic Challenges, and the Potential of Biomarker-Based Approaches Charles M. Manyelo,a Regan S. Solomons,b Gerhard Walzl,a and Novel N. Chegou a (Published online 2021 Feb 18. Prepublished online 2020 Oct 21. doi: 10.1128/JCM.01771-20). This review focuses on many similar aspects of the authors current document and would enrich the discussion in the very least. Repeating same arguments should be avoided.

Authors: Thank you for the suggestions.  These references have now been included.

2 - Other algorithms for diagnosis have been reported, and are not referenced.

Authors: This has now been added.

3 - The section of ‘Natural History and Pathogenesis’ lacks a more thorough explanation of the immune response during the infection.

Authors: Thank you for this comment. We have now added substantial additional content around the immune response seen in TBM.  

4- Both figure 1 and 2 are very helpful for the discussion and interpretation of the authors key points. However, a more detailed examination in the text should be included. This is an area that distinguishes the paper and should be exploited and expanded

Authors: Thank you for raising this. Additions have now been made to epidemiology and diagnosis sections where these figures are referenced to explain the figures more completely.

Minor issues -

1 - The authors correctly point out that prompt diagnosis and reporting are important factors in successful treatment of TBM, however no specific recommendation or research is proposed.

Authors- This has now been added to Table 1.

2 - In the text the authors correctly state that Xpert is not sensitive enough for TBM, but it is added in the Figure 2. Please explain. If Xpert is not recommended, what would the authors suggest.

Authors- The development of Xpert MTB/RIF was a significant step forward for improving the diagnosis of TB and the detection of rifampicin resistance globally such that it is now recommended by WHO as the initial diagnostic test in both pulmonary and extrapulmonary TB. In line with this recommendation, we have included Xpert MTB/RIF or Xpert Ultra in Figure 2.  

However, It also important we highlight in our manuscript the limitations of Xpert in diagnosing paucibacillary disease, and that sensitivity is still suboptimal in TBM.  Though there has been much research activity evaluating new rapid diagnostic tests/approaches in TBM none have adequate evidence for routine clinical use.  This crucial area of research has been highlighted in table 1.  

See amendment to legend of figure 2 for clarification.

3- A key point would be to identify and validate biomarkers, as the authors state in the text.

Authors: We agree with the reviewer on this point and have added an additional reference to highlight that not only are biomarkers needed but ones that could be used as point of care tests. 

Are there any genetic predispositions for TBM?

Authors: there are limited data to evaluate genetic susceptibility to TBM however. Studies suggest that the leukotreine A4 hyroxylase (LTA4H) may influence the host inflammatory response in certain populations with TBM, however this research is ongoing and LTA4H is only one of many genes which has yet to be fully studied in TBM. We have expanded on this in the text and added content to Table 1.

How does malnutrition factor into the prognosis?

Malnutrition is widely accepted to be a risk factor for poor outcome in childhood tuberculosis and is likely to be a risk factor for poor outcome in TBM. However, few studies of outcome in child TBM have evaluated the role of nutrition in TBM prognosis so we have removed the reference to malnutrition being a risk factor for poor outcome in TBM.

Is there an estimate on how long between pulmonary TB infection and TBM development?

A review of the pre-chemotherapy literature has shown that the time from TB infection to the development of TBM development in the high risk age group of <2 years to be within 12 months. We have highlighted this in the natural history and pathogenesis section: ‘A review of the natural history of childhood intrathoracic tuberculosis in the pre-chemotherapy era found that young age (<2 years of age) was the major determinant of progression from TB infection to disease; pulmonary disease developed in 30-40% and TBM or miliary disease in 10-20% of infants, and the highest risk was within 4 months of infection.’

Reviewer 3 Report

Congratulations on this comprehensive review of such a relevant subject. The authors made an excellent job summarizing the most relevant evidence currently available. Suggestions for future research are sound. I have very few suggestions.

I would advice the authors to cite a previous meta-analyses on the role of BCG to prevent TBM (Colditz GA et al. Efficacy of BCG vaccine in the prevention of tuberculosis. Meta-analysis of the published literature. JAMA. 1994 Mar 2;271(9):698-702. PMID: 8309034), along with the one already cited.

Please revise phrasing. Passages such as "in whom TBM or its treatment can impact their developing brains" need rewording. "It is challenging to diagnose and treat and much of the damage that occurs has already taken place before the child is started on appropriate therapy." could be replaced by "It is challenging to diagnose and treat early; much of the damage has already occurred before the child is started on appropriate therapy." Also, avoid repeated  "however" in the middle of the sentence. Many commas are missing. In brief, minor but careful editorial review is needed, I have only highlighted a couple of examples. 

A minor editorial remark: the quality of Figure 2 needs improvement for better readability.

Author Response

Congratulations on this comprehensive review of such a relevant subject. The authors made an excellent job summarizing the most relevant evidence currently available. Suggestions for future research are sound. I have very few suggestions.

I would advice the authors to cite a previous meta-analyses on the role of BCG to prevent TBM (Colditz GA et al. Efficacy of BCG vaccine in the prevention of tuberculosis. Meta-analysis of the published literature. JAMA. 1994 Mar 2;271(9):698-702. PMID: 8309034), along with the one already cited.

Authors: Thank you for this suggestion. This citation has now been included.

Please revise phrasing. Passages such as "in whom TBM or its treatment can impact their developing brains" need rewording. "It is challenging to diagnose and treat and much of the damage that occurs has already taken place before the child is started on appropriate therapy." could be replaced by "It is challenging to diagnose and treat early; much of the damage has already occurred before the child is started on appropriate therapy." Also, avoid repeated  "however" in the middle of the sentence. Many commas are missing. In brief, minor but careful editorial review is needed, I have only highlighted a couple of examples.

Authors: We have cut the first phrase highlighted and then made the second suggested change. The use of however in the middle of sentences have been altered so that sentences start with although or while. We have gone through and critically edited for style and punctuation. We are happy for the journal editors to make further editorial changes should they deem them appropriate.

A minor editorial remark: the quality of Figure 2 needs improvement for better readability.

Authors: The Figures were developed in PowerPoint and then converted to JPEG at a resolution of 300 dpi x 300 dpi. This is a very high resolution and easily enough for publication quality. However, on reviewing the PDF that is generated by the online system for peer review it is noted that the resolution is poor. We will work with the editorial team to confirm that when the article is typeset, the images will be of the highest possible resolution.

Round 2

Reviewer 2 Report

The authors significantly improved the document and I recommend it for publication.